# AGE-MORL: Agent-Guided Evolutionary Control for Multi-Objective Reinforcement Learning

## Abstract

strategies, in terms of both solution quality and stability. Multi-objective reinforcement learning (MORL) provides a powerful paradigm for solving problems with conflicting objectives. However, their performance can be highly dependent on the selection of search operators, which often follows predefined and non-adaptive strategies. In this paper, we propose a novel hierarchical framework, termed Agent-Guided Evolutionary Control framework for Multi-Objective Reinforcement Learning (AGE-MORL), which leverages Deep Reinforcement Learning (DRL) to adaptively select operators for multi-objective optimization algorithms at a high level. The operator selection problem is formulated as a Markov Decision Process (MDP), where a reinforcement learning agent learns a dynamic strategy-selection policy based on the evolving state of the optimization process. To enhance search effectiveness, we design a set of intelligent search operators based on geometric analysis of the Pareto front, including a sector-based exploration mechanism to explore sparse regions. Further, to enhance population diversity and escape local optima, we integrate a probabilistic acceptance model that combines a Simulated Annealing (SA) criterion with a Blink mechanism. Experiments on a diverse set of multi-objective optimization problems show that the proposed method significantly outperforms existing state-of-the-art methods, in terms of both solution quality and stability.

## 1 Introduction

Deep Reinforcement Learning (DRL) Arulkumaran et al. (2017) has emerged as a powerful paradigm for solving complex sequential decision-making problems, and it has achieved remarkable success in domains like robotics Gu et al. (2017), resource allocation Zhao & Zhao (2021), and autonomous driving Tang et al. (2025). In standard RL, an agent's learning is driven by a scalar reward function that defines the sole optimization objective. However, many real-world applications are inherently multi-faceted, presenting agents with several, often conflicting, objectives Siddique et al. (2020). For instance, a logistics robot in a warehouse must simultaneously maximize delivery speed to meet deadlines while minimizing energy consumption to extend its operational life. In such multi-objective decision-making problems, a single optimal policy does not exist, as it is impossible to simultaneously maximize all conflicting objectives. Instead, the solution is a set of Pareto-optimal policies. Adapting to specific requirements and operational contexts, a human user or a higher-level system must select a trade-off that reflects their preferences. Each distinct preference corresponds to a different policy within the Pareto-optimal set, allowing for flexible and context-aware decision-making.

To deal with these challenges, researchers have paid more attention to Multi-Objective Reinforcement Learning (MORL) and many advanced approaches have been constructed. The conventional approach, known as single-policy MORL, transform the multi-dimensional objective space into a scalar representation through the static assignment of weights Van Moffaert et al. (2013). This scalarization strategy, however, is sensitive to the relative scales of the objectives and requires specialized expertise to define a meaningful weight configuration Nguyen et al. (2020). Crucially, this yields only a single trade-off solution. Discovering the entire Pareto front would require numerous, computationally expensive retraining cycles, which is impractical for most applications.

To address these limitations, multi-policy methods have been proposed, which aim to learn a set of policies covering multiple preferences in a single training run. These approaches encompass several distinct strategies. One line of work extends classic value-based algorithms, such as Q-learning, to maintain and update a set of non-dominated value vectors for each state Yang et al. (2019); de Oliveira et al. (2021). Another prominent strategy involves training a single, preference-conditioned network that takes a desired trade-off as an input to generate the corresponding policy Su et al. (2024); Huang et al. (2022). A third major paradigm, often operating within an evolutionary framework, maintains a population of policies and leverages mechanisms like decomposition or meta-learning to guide their collective search towards the Pareto front Chu et al. (2020); Shen et al. (2020); Hu & Luo (2024).

However, while mathematically elegant, the fixed update logic of these methods exposes critical limitations in global search adaptation. First, their optimization strategies are typically static and non-adaptive, relying on a single, pre-defined rule that may not be suitable for all problem types or across different stages of the search. This rigidity often necessitates manual, offline parameter tuning, creating a high dependency on domain expertise. Second, their decision-making is fundamentally localized, focusing on improving individual policies or their immediate neighbors, rather than leveraging a holistic, front-level perspective to identify and target diverse regions for optimization. Finally, this localized view, combined with a lack of robust exploration mechanisms, makes them highly susceptible to premature convergence, where the entire search stagnates in a suboptimal region of the Pareto front.

To address these challenges, we propose a novel hierarchical control framework, termed Agent-Guided Evolutionary Control for MORL (AGE-MORL). We model the high-level selection of search operators as a Markov Decision Process (MDP), and train a high-level RL agent to act as an adaptive controller. This agent learns a dynamic policy to choose the most suitable search strategy based on a comprehensive state representation of the search, allowing it to systemically balance exploration and exploitation. The main contributions of this paper are as follows:

1. We propose AGE-MORL, a novel hierarchical control framework that rigorously formulates the adaptive operator selection problem in MOEAs as a Markov Decision Process (MDP). By training a high level agent, our framework learns a dynamic policy to intelligently guide the evolutionary search at a systemic level, overcoming the limitations of predefined strategies by making real-time adjustments based on the evolving search state.

2. We design a set of intelligent, geometry-aware search operators that analyze the Pareto front to dynamically distinguish between dense and sparse regions. This enables targeted strategies, balancing intensive exploitation with directed exploration to enhance search efficiency and the quality of the final solution set.

3. We introduce a synergistic mechanism combining a Blink mechanism with an adaptive Simulated Annealing (SA) criterion to prevent premature convergence. This hybrid approach promotes population diversity by disrupting elite dominance and allows the search to escape local optima by probabilistically accepting non-improving solutions, ensuring robust global exploration.

4. We conduct extensive experiments on a diverse set of multi-objective combinatorial optimization problems. The results demonstrate that AGE-MORL significantly outperforms traditional MOEAs and state-of-the-art baselines in terms of both solution quality and stability.

## 2    RELATED WORK

### 2.1    MORL PROBLEM

A multi-objective sequential decision problem is formulated as a Multi-Objective Markov Decision Process (MOMDP) Hayes et al. (2022), defined by the tuple $\langle S, A, T, \gamma, \mu, \mathbf{R} \rangle$. Here, $S$ is the state space, $A$ is the action space, $T : S \times A \times S \rightarrow [0, 1]$ is the probabilistic transition function, and $\mu : S \rightarrow [0, 1]$ is the initial state distribution.

The key multi-objective components are the vector-valued discount factor $\gamma = [\gamma_1, \ldots, \gamma_m]^\top \in [0, 1]^m$ and the vector-valued reward function $\mathbf{R} = [r_1, \ldots, r_m]^T : S \times A \times S \rightarrow \mathbb{R}^m$. These vectors account for $m$ distinct objectives, where $m \geq 2$. In this framework, an agent operates under a policy $\pi_\theta \in \Pi$, where $\Pi$ is the set of all possible policies. A policy is a mapping $\pi_\theta : S \rightarrow A$., which maps states to actions. The performance of a policy is evaluated by its corresponding vector of expected returns, $\mathbf{F}(\pi) = \mathbf{J}^\pi = [J_1^\pi, \ldots, J_m^\pi]^\top$, where each component $J_i^\pi$ is given by:

$$J_i^\pi = \mathbb{E}\left[ \sum_{k=0}^{\infty} \gamma_i^k r_i(s_k, a_k, s_{k+1}) \mid \pi, \mu \right]. \tag{1}$$

The MORL problem could be expressed as:

$$\max_{\pi} \mathbb{F}(\pi) = \max_{\pi} [J_1^\pi, J_2^\pi, \ldots, J_m^\pi]^T. \tag{2}$$

Unlike in single-objective settings where policies can be clearly ordered, MORL problems involve conflicting objectives. Consequently, a single policy $\pi^*$ that simultaneously maximizes all objectives typically does not exist. This necessitates the use of the following Pareto concepts to compare policies Cai et al. (2023); Qian et al. (2019).

**Pareto Dominance:** A policy $\pi$ is said to dominate another policy $\pi'$ (denoted $\pi \succ \pi'$) if it is at least as good on all objectives and strictly better on at least one. Formally, $\pi \succ \pi'$ if and only if $J_i^\pi \geq J_i^{\pi'}$ for all $i \in \{1, \ldots, m\}$ and there exists some $j$ such that $J_j^\pi > J_j^{\pi'}$.

**Pareto Optimality:** A policy $\pi^*$ is *Pareto optimal* if no other policy in the policy space $\Pi$ dominates it. In other words, there is no $\hat{\pi} \in \Pi$ such that $\hat{\pi} \succ \pi^*$.

**Pareto Set and Front:** The set of all Pareto optimal policies forms the Pareto set. The image of this set in the objective space is known as the Pareto front.

A common method to find a single Pareto optimal solution is to scalarize the objective vector using a utility function, such as a linear weighted sum $u(\mathbf{J}^\pi, \omega) = \omega^\top \mathbf{J}^\pi$ with a preference vector $\omega$, where $\omega = [\omega_1, \ldots, \omega_m]^T, \omega_i \geq 0$ and $\sum_{i=1}^m \omega_i = 1$. However, this only yields a single solution for a given $\omega$. Since the Pareto front can be extremely large in complex problems, the primary goal of MORL is to efficiently find a good approximation of the entire Pareto front.

## 2.2 MORL ALGORITHMS

While standard reinforcement learning excels at single-objective tasks, many real-world scenarios require balancing multiple, conflicting objectives simultaneously. MORL provides a powerful framework for this challenge by explicitly handling the trade-offs between different goals. Existing MORL methods are broadly classified into two primary categories: single-policy and multi-policy approaches, which we review below.

### 2.2.1 SINGLE-POLICY METHODS

Single-policy methods aim to simplify the multi-objective problem by transforming it into a conventional single-objective task. The most prevalent technique within this category is scalarization, where the multi-objective reward vector is converted into a single scalar reward using a predefined preference or weight vector. An agent then employs standard single-objective RL algorithms to learn a policy that optimizes this scalarized reward. Khamis et al. proposed a multi-agent framework that scalarizes multiple performance indices, such as minimizing waiting time and maximizing flow, into a unified objective to guide the learning of cooperative traffic controllers Khamis & Gomaa (2014). Duan et al., in their influential benchmarking work, created multi-objective control tasks by linearly combining rewards for competing goals, such as forward speed and energy efficiency, into a single scalar value for standard RL algorithms to optimize Duan et al. (2016). To address the challenge of setting preferences across different scales, Abdolmaleki et al. introduced MO-MPO, an algorithm that enables practitioners to specify desired trade-offs in a scale-invariant manner by learning and combining action distributions for each objective Abdolmaleki et al. (2020).

Despite their simplicity and ease of implementation, these methods are constrained by significant limitations. Firstly, they typically require a priori specification of user preferences (e.g., the weight vector), which can be non-intuitive or difficult to articulate accurately for complex tasks. Secondly, and more critically, if the desired trade-off or application scenario changes, the learned policy often becomes suboptimal, necessitating a full-scale retraining process. This inflexibility makes single-policy methods ill-suited for applications where a user may need to explore various trade-off solutions or adapt to new requirements dynamically.

### 2.2.2 MULTI-POLICY METHODS

Multi-policy methods address the limitations of their single-policy counterparts by aiming to learn an entire set of policies that approximate the Pareto front in a single training run. This provides users with a diverse range of trade-off solutions to choose from, offering significantly greater flexibility. Early approaches in this area often involved running a single-policy algorithm over various preferences, which could be computationally inefficient Mossalam et al. (2016); Zuluaga et al. (2016). To avoid this explicit iterative search, other methods directly learn a set of solutions. Some studies focus on multi-objective Q-learning methods, where a single network architecture acquires a set of policies by taking preferences as an input and applying vectorized value function updates. Abels et al. (2019); Yang et al. (2019). Chen et al. introduced a novel meta-learning approach for MORL, training an initial meta-policy that is subsequently fine-tuned for various preferences to efficiently generate a superior Pareto front Chen et al. (2019). PGMORL employs an evolutionary framework to select policies based on a predictive model, subsequently updating them using the Proximal Policy Optimization (PPO) algorithm Xu et al. (2020). Pareto Ascent Directional Decomposition MORL (PA2DMORL) Hu & Luo (2024) constructs an evolutionary framework where each policy is updated along a mathematically computed Pareto ascent direction to achieve simultaneous improvement on all objectives.

However, while mathematically elegant, these methods are constrained by a fixed, pre-defined update logic. This rigidity presents a major obstacle, as they lack a higher-level mechanism to adapt their global search strategy. Their decision-making is fundamentally localized, preventing them from analyzing the holistic state of the Pareto front to, for instance, distinguish sparse regions from dense ones. Consequently, without robust and adaptive exploration strategies, they are prone to premature convergence and search stagnation.

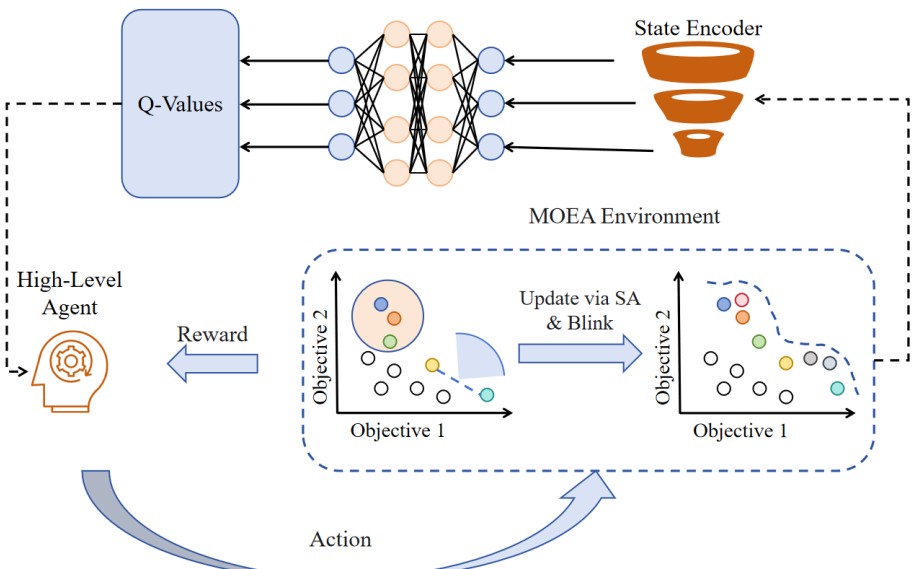

Figure 1: Illustration of the proposed AGE-MORL. The high-level DRL agent learns a policy to select an appropriate action, guiding the low-level MOEA. The MOEA Environment visualizes two key search strategies from our intelligent operator library: a circular neighborhood search to exploit dense regions and a sector-based search to explore sparse regions. The population update is enhanced by our synergistic SA and Blink mechanism.

This motivates our work to move beyond such static, localized approaches. We propose a hierarchical control framework where a high-level agent learns to dynamically select search strategies from a diverse operator library. By observing the global state of the search, our framework can intelligently balance exploration and exploitation on a systemic level—a capability not present in methods that rely on a fixed update rule.

## 3 METHOD

### 3.1 THE OVERALL FRAMEWORK

Our proposed method, AGE-MORL, is motivated by the need to overcome three fundamental challenges in multi-policy MORL: the reliance on static strategies that require expert tuning, decision-making based on a localized perspective, and a high susceptibility to premature convergence. Traditional methods, with their fixed update rules, lack the strategic flexibility to adapt to the evolving search landscape. We address these limitations by introducing a hierarchical control framework that reframes the global search adaptation challenge as a learning problem, enabling an agent to dynamically learn and execute a high-level search strategy.

As depicted in Figure 1, the AGE-MORL framework consists of two interacting levels: a high-level DRL agent acting as an adaptive controller, and a low-level MOEA environment that maintains the solution population. This interaction is formalized as a MDP. At each stage, the high-level agent observes a comprehensive state of the MOEA, capturing a holistic view of the entire Pareto front. Based on this state, it leverages its learned policy to select the most appropriate search operator (action) from a diverse library. The MOEA executes this operator, and the resulting change in search performance provides a reward signal to the agent. Through this closed-loop process, the agent's policy is continuously optimized to maximize cumulative rewards, ensuring the entire search is intelligently and cohesively guided towards a high-quality, well-distributed set of solutions. The key components of this MDP formulation are detailed below.

State Representation ($\mathcal{S}$): To enable informed decision-making, the high-level agent requires a comprehensive yet compact representation of the MOEA's current state. We engineer a 10-dimensional state vector $s_t$ that encapsulates the search dynamics across three key axes: performance, diversity, and search condition. This includes performance metrics such as the recent trend and stability of the Hypervolume (HV); diversity indicators like the variance of solutions in the objective space and overall front coverage; and search condition flags that detect stagnation. This engineered state representation provides a rich, low-dimensional snapshot of the search, allowing the agent to discern

complex scenarios like premature convergence or an unevenly explored front. A formal definition of each component in the state vector is detailed in the Appendix.

Action Space ($\mathcal{A}$): The action space $\mathcal{A}$ constitutes a discrete library of high-level search strategies, each tailored to address a specific evolutionary challenge. This library includes operators designed to, for example, intensify the search within promising, high-density regions of the Pareto front, or alternatively, to expand the search into sparsely-populated gaps to improve front coverage. The selection of an appropriate strategy is informed by a geometric analysis of the current front. Furthermore, the execution of any chosen operator is enhanced by advanced mechanisms, such as a Blink mechanism to maintain diversity and an adaptive Simulated Annealing criterion for global exploration, which are detailed in the subsequent section.

Reward Function ($\mathcal{R}$): The reward function is critical for guiding the agent towards beneficial long-term strategies. The primary reward signal, $r_t$, is directly tied to the relative improvement in the hypervolume (HV) of the Pareto front. To encourage more nuanced behaviors, this base reward is augmented with situational bonuses and penalties. For instance, a positive bonus is awarded for selecting an exploratory action during a detected period of stagnation, while a penalty is applied for choosing an overly aggressive convergence strategy early in the search. This multi-faceted reward structure, formally $r_t = f(HV_t, HV_{t-1}, s_t, a_t)$, ensures the agent learns to optimize for both immediate performance and long-term search health.

The entire interactive process is formalized in Algorithm 1. It details the main loop where the high level agent observes the state, selects a strategic action, and updates its Q-network based on the received reward. The algorithm also highlights how our core mechanisms are integrated into the evolutionary cycle.

---

**Algorithm 1** AGE-MORL Guided MOEA

---

**Require:** MOEA parameters (population size $N$); AGE-MORL parameters ($\alpha, \gamma, \epsilon$)
**Require:** Total number of generations $T_{\max}$
**Ensure:** Final Pareto optimal set $PF_{\text{final}}$
 1: Initialize MOEA population $P_0$ and global SA temperature $T_0$
 2: Initialize Q-network $Q_\theta$ and target network $Q_{\theta'}$, replay buffer $\mathcal{D}$
 3: $s_0 \leftarrow$ EncodeState($P_0$)
 4: **for** $t = 0$ to $T_{\max} - 1$ **do**
 5:     *// Meta-level decision making*
 6:     Select action $a_t \sim \pi(s_t)$ (e.g., $\epsilon$-greedy from $Q_\theta(s_t)$)
 7:     *// High-level strategy execution*
 8:     $\mathcal{I}_{\text{candidate}} \leftarrow$ SelectIndividualsByGeometricSearch($P_t, a_t$)
 9:     $\mathcal{I}_{\text{selected}} \leftarrow$ ApplyBlinkMechanism($\mathcal{I}_{\text{candidate}}, a_t$)
10:     *// Low-level evolution with adaptive SA*
11:     $T_{\text{local}} \leftarrow$ AdaptSAt-Temperature($T_t, a_t$)
12:     $P_{t+1} \leftarrow$ EvolveSelectedIndividuals($P_t, \mathcal{I}_{\text{selected}}, T_{\text{local}}$)
13:     *// Feedback and learning*
14:     $s_{t+1} \leftarrow$ EncodeState($P_{t+1}$)
15:     $r_t \leftarrow$ CalculateReward($P_t, P_{t+1}, a_t$)
16:     Store transition $(s_t, a_t, r_t, s_{t+1})$ in $\mathcal{D}$
17:     **if** $|\mathcal{D}| >$ batch_size **then**
18:         Sample and update Q-network $Q_\theta$ using target network $Q_{\theta'}$
19:     **end if**
20:     Update global SA temperature $T_{t+1} \leftarrow$ cool($T_t$)
21: **end for**
22: **return** Extract final Pareto set $PF_{\text{final}}$ from $P_{T_{\max}}$

---

## 3.2 GEOMETRY-AWARE SEARCH STRATEGIES

Standard evolutionary operators often treat all solutions uniformly, lacking the awareness to dynamically allocate search effort where it is most needed. To overcome this, we introduce a suite of intelligent search strategies that leverage geometric information from the current Pareto front to explicitly manage the trade-off between exploitation and exploration. These strategies are the core components of the action space $\mathcal{A}$ available to the AGE-MORL agent.

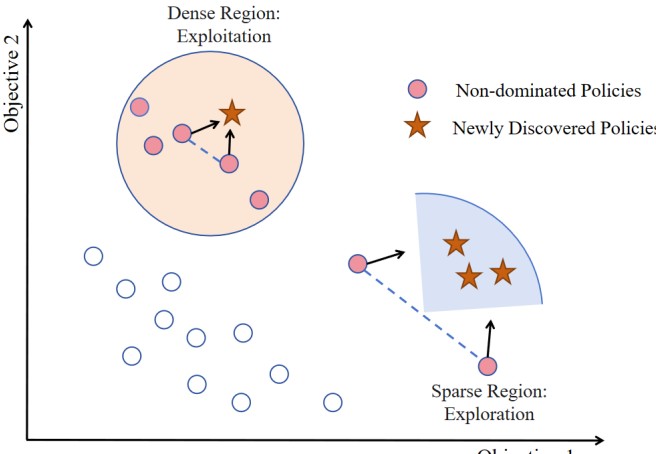

Figure 2: Differentiated Search Strategies. For dense regions, a circular search intensifies local exploitation. For sparse regions, a sector-based search explores uncharted areas to expand the front.

### 3.2.1 GEOMETRIC REGION IDENTIFICATION

The first step is to analyze the current Pareto front approximation, $\mathcal{PF}$, to identify regions that are either densely or sparsely populated. We compute the set of all pairwise Euclidean distances, $D = \{\|\mathbf{o}_i - \mathbf{o}_j\|_2 \mid p_i, p_j \in \mathcal{PF}, i \neq j\}$. From this set, we determine the minimum and maximum distances, $\delta_{\min} = \min(D)$ and $\delta_{\max} = \max(D)$. Using these values, we formally define the set of dense pairs, $\mathcal{P}_{\text{dense}}$, and the set of sparse pairs, $\mathcal{P}_{\text{sparse}}$, as follows:

$$\mathcal{P}_{\text{dense}} = \{(p_i, p_j) \mid \|\mathbf{o}_i - \mathbf{o}_j\|_2 \leq c_d \cdot \delta_{\min}\} \tag{3}$$

$$\mathcal{P}_{\text{sparse}} = \{(p_i, p_j) \mid \|\mathbf{o}_i - \mathbf{o}_j\|_2 \geq c_s \cdot \delta_{\max}\} \tag{4}$$

where $c_d > 1$ and $c_s < 1$ are threshold scaling factors. The pairs in $\mathcal{P}_{\text{dense}}$ become candidates for intensive exploitation, while those in $\mathcal{P}_{\text{sparse}}$ mark gaps in the front that are primary targets for exploration.

### 3.2.2 DIFFERENTIATED NEIGHBORHOOD SEARCH

To act upon the identified regions, we employ differentiated search patterns. For dense regions, a circular neighborhood search is initiated to intensify the search locally; its radius is scaled by a factor $r$ relative to the average distance on the front. For sparse regions, a more targeted sector-based search is employed. Its geometry is controlled by a length factor $l$ for search depth and a width factor $w$ for breadth, while its vector-based implementation allows for either a focused half scan or a comprehensive full scan (using absolute projection) to explore uncharted territories.

## 3.3 SYNERGISTIC MECHANISM FOR EXPLORATION AND ACCEPTANCE

We introduce a synergistic mechanism that combines a high-level Blink mechanism with a low-level adaptive Simulated Annealing (SA) criterion to enhance search robustness.

### 3.3.1 HIGH-LEVEL DIVERSITY PRESERVATION VIA BLINK MECHANISM

The Blink mechanism is a high-level strategy applied before a selected search action is executed. It preserves diversity by probabilistically excluding the top-performing individuals from the candidate set, thus preventing premature convergence. The exclusion probability is context-dependent, set higher for exploration actions than for exploitation ones to adaptively manage the diversity pressure.

### 3.3.2 LOW-LEVEL ACCEPTANCE VIA ADAPTIVE SIMULATED ANNEALING

At the low-level, an adaptive SA criterion governs the acceptance of new solutions. This allows the search to escape local optima by accepting a non-improving solution (where the scalarized objective change $\Delta E < 0$) with the

Metropolis probability:

$$P_{\text{accept}} = \exp\left(\frac{\Delta E}{T}\right) \tag{5}$$

The synergy lies in adaptively modulating the temperature $T$ based on the agent's high-level action. Exploratory actions increase $T$ to encourage probing new regions, while exploitative actions decrease $T$ for stricter refinement. This dynamically aligns the low-level acceptance threshold with the high-level strategic goal.

## 4 EXPERIMENTS

### 4.1 EVALUATION METRICS

To evaluate the performance of our proposed framework, we employ two widely-recognized metrics: HV and Inverted Generational Distance (IGD) Yao et al. (2025).

1) Hypervolume (HV): The HV metric provides a comprehensive assessment of both the convergence and diversity of an approximated Pareto front, without requiring a known ground-truth front. It calculates the volume of the objective space dominated by the solution set relative to a predefined reference point. A key advantage of HV is its strict Pareto compliance; thus, a larger HV value consistently indicates a superior overall performance.

2) Inverted Generational Distance (IGD): The IGD metric measures the quality of the generated front by evaluating its proximity to a reference Pareto front. For a fair comparison, the reference set in our experiments is constructed from the non-dominated union of all solutions found by the union of all generated heuristics. A lower IGD value is preferred, signifying that the generated solutions are closer to the reference set and indicating better performance in both convergence and diversity.

The detailed mathematical formulations for both metrics can be found in the Appendix.

### 4.2 SIMULATION ENVIRONMENT

We evaluate our proposed framework on a diverse set of three challenging multi-objective combinatorial optimization problems: the Traveling Salesman Problem (TSP), the Capacitated Vehicle Routing Problem (CVRP), and the Pickup and Delivery TSP (PDSTSP). To better reflect the unpredictability of real-world scenarios, we model all environments with uncertain travel times, making the optimization process more dynamic and challenging. The traffic volume data is obtained from Transportation networks for research core team (2023). A common primary objective across all these tasks is the minimization of the total travel distance, while the second, conflicting objective varies to test our method's adaptability. For the TSP, this second objective is another distance metric calculated from independent coordinates, creating a direct conflict between two notions of path length Li et al. (2020). For the CVRP, it minimizes aggregate customer waiting time (e.g., the sum or maximum of waiting/arrival delays), introducing a trade-off between timely service and total travel distance. Finally, for the PDSTSP, it minimizes total operational cost with mode-dependent rates for truck and drone, which can conflict with the shortest-path objective when cheaper modes require longer or coordinated routes. In these sequential decision-making tasks, the state is represented by the problem instance alongside the current partial solution, which includes unserved nodes, each vehicle's location and remaining capacity, and vehicle-specific status (e.g., drone battery). The agent's action, in turn, is to select the next feasible node to visit, determining its assignment to a specific vehicle and its position in the tour, subject only to capacity and reachability constraints. For each problem type, we create instances with 50, 100, and 200 nodes to evaluate the method's performance across different scales.

To ensure a fair and consistent comparison, all experiments were conducted on a unified platform equipped with an Intel Core i9-14900KF processor, 32GB of memory, and an NVIDIA GeForce RTX 4080 GPU.

### 4.3 BASELINE

We compare our proposed method, against its ablated version and three other baseline algorithms. The ablated version, denoted as AGE-MORL-A, removes the high level agent and operates with a fixed, predefined search strategy. In addition, we benchmark our method against three prominent multi-objective algorithms: PA2D-MORL Hu & Luo (2024), MOEA/D Zhang & Li (2007), and PFA Parisi et al. (2014). PA2D-MORL is selected as a state-of-the-art multi-policy MORL algorithm that represents a different paradigm for policy improvement. MOEA/D and PFA are chosen due to their demonstrated strong performance in numerous multi-objective optimization benchmarks. Further details about the implementation of these baselines are provided in the Appendix.

For all experiments, every algorithm, including the baselines and our proposed method, operates under identical experimental conditions. This includes using the same population size, underlying policy update rules for low-level agents, and the same evaluation environments and reward structures, ensuring a rigorous and fair assessment. Detailed parameter settings are presented in the Appendix.

### 4.4 RESULTS

Table 1 presents the quantitative evaluation results across all environments, showing the final HV and IGD scores. Figure 3 illustrates the HV convergence curves over 1000 iterations and provides a visual comparison of the final Pareto front approximations obtained by each algorithm. All reported results are averaged over multiple independent runs.

Across all problem instances and sizes, AGE-MORL achieves the best performance in both HV and IGD metrics. Its superior HV scores and lower IGD values demonstrate that it generates higher-quality, better-converged Pareto fronts. Notably, for several environments such as $CVRP_{50}$ and $PDSTSP_{100}$, AGE-MORL achieves an IGD score of 0.000, which means its generated front completely dominates the combined results of all other methods, as the reference front is formed exclusively by its solutions. Furthermore, the convergence curves in Figure 3 show that AGE-MORL not only reaches a higher final HV but also exhibits a faster and more stable convergence rate compared to all competitors.

The baseline algorithms exhibit clear limitations in comparison. As shown in Figure 3, PA2D and PFA, while effective on smaller instances like $TSP_{50}$, struggle to scale, often converging to solutions that are clearly dominated by those from AGE-MORL on larger problems. MOEA/D tends to find a concentrated cluster of solutions but consistently fails to capture the full extent of the Pareto front, particularly in the extreme regions. This lack of diversity is reflected in its consistently high IGD values.

Our ablation study further highlights the importance of the adaptive high level agent. While the ablated version, AGE-MORL-A, outperforms most external baselines, it is consistently surpassed by the full AGE-MORL framework. This performance gap, evident in both Table 1 and Figure 3, demonstrates that the intelligent, state-aware strategy selection is essential for achieving superior results by more effectively navigating the optimization landscape.

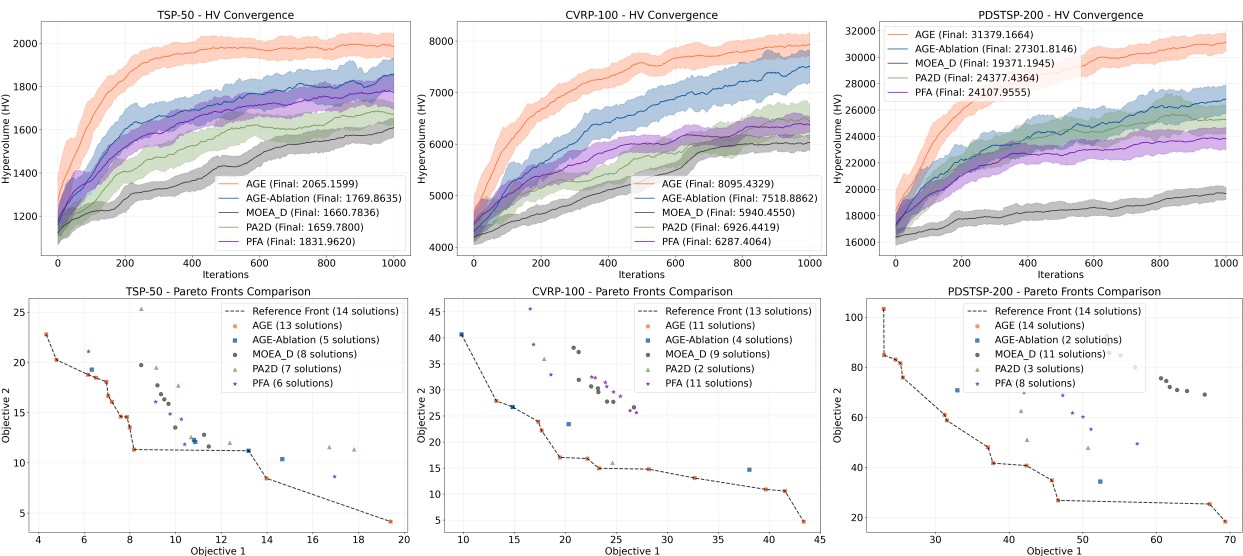

Figure 3: Performance of AGE-MORL against baselines on multi-objective combinatorial optimization tasks. The plots show HV convergence and final Pareto fronts across different tasks (TSP-50, CVRP-100, PDSTSP-200), AGE-MORL achieves a significantly higher-quality and more diverse set of solutions.

To further understand its superior performance, we analyze the dynamic policy learned by AGE-MORL's high-level agent. Figure 4 illustrates that the agent learns a highly adaptive policy, rather than relying on any single strategy. The usage probabilities of different strategies shift significantly throughout the search, and the overall pattern differs notably between the TSP and CVRP tasks. This demonstrates that the agent's policy is both stage-dependent and

Table 1: Evaluation results for all algorithms. The experiments are based on three problem types at three different scales. Both the average HV and IGD metrics are reported. All data are based on 10 independent runs.

| Env | AGE-MORL | | AGE-MORL-A | | PA2D | | PFA | | MOEA/D | |
|---|---|---|---|---|---|---|---|---|---|---|
| | HV ↑ | IGD ↓ | HV ↑ | IGD ↓ | HV ↑ | IGD ↓ | HV ↑ | IGD ↓ | HV ↑ | IGD ↓ |
| TSP$_{50}$ | **2065.160** | **0.204** | 1769.864 | 2.715 | 1659.780 | 3.538 | 1831.962 | 2.540 | 1660.784 | 3.352 |
| TSP$_{100}$ | **7863.771** | **0.136** | 7086.993 | 6.965 | 6547.127 | 8.236 | 7000.269 | 7.550 | 5902.952 | 11.118 |
| TSP$_{200}$ | **29387.024** | **0.837** | 28593.846 | 6.398 | 24499.597 | 18.037 | 25246.140 | 15.002 | 22279.707 | 25.784 |
| CVRP$_{50}$ | **2069.342** | **0.000** | 1830.794 | 3.717 | 1674.904 | 5.384 | 1797.222 | 4.017 | 1543.497 | 6.619 |
| CVRP$_{100}$ | **8095.432** | **1.168** | 7518.89 | 5.115 | 6926.44 | 9.689 | 6287.40 | 12.684 | 5940.455 | 13.559 |
| CVRP$_{200}$ | **31052.356** | **0.226** | 28181.431 | 12.731 | 24038.229 | 23.191 | 25203.703 | 18.997 | 21641.741 | 27.532 |
| PDSTSP$_{50}$ | **2119.719** | **0.454** | 1774.347 | 3.413 | 1814.824 | 5.626 | 1704.631 | 6.414 | 1478.394 | 7.768 |
| PDSTSP$_{100}$ | **8558.986** | **0.000** | 7174.489 | 7.178 | 6536.119 | 11.239 | 6184.068 | 12.305 | 5450.404 | 19.121 |
| PDSTSP$_{200}$ | **31379.166** | **0.000** | 27301.814 | 15.421 | 24377.436 | 20.905 | 24107.955 | 19.515 | 19371.194 | 36.142 |

problem-dependent, confirming that AGE-MORL's strength lies in its intelligent, real-time adaptation of the search strategy.

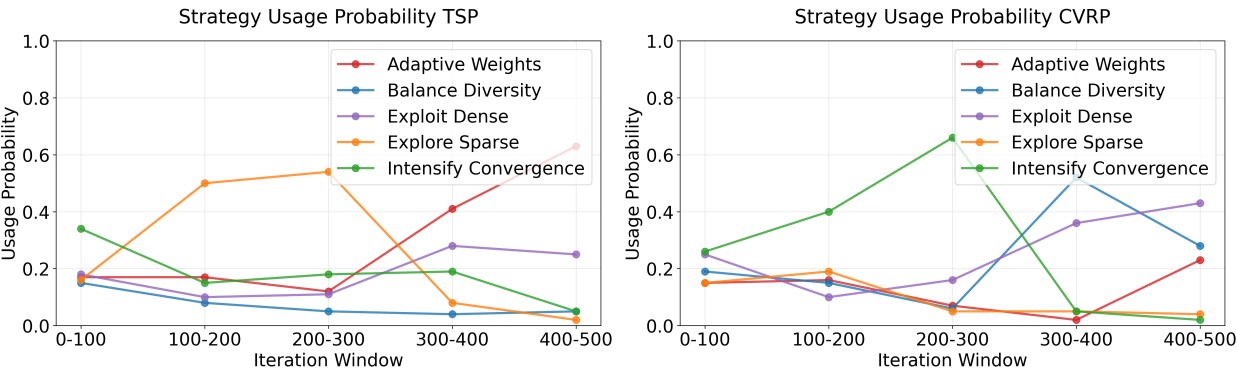

Figure 4: Dynamic strategy usage probabilities for the AGE-MORL agent on TSP and CVRP. The figures show the probability of selecting each search strategy within different iteration windows.

## 5 CONCLUSION

In this paper, we proposed AGE-MORL, a novel hierarchical control framework for intelligently guiding the search process of MOEAs in solving complex multi-objective optimization problems. By formulating operator selection as a Markov Decision Process, we train a high-level RL agent to act as an adaptive controller. This agent learns a dynamic policy to intelligently select from a suite of geometry-aware search operators and leverages a synergistic exploration mechanism to effectively balance exploration and exploitation. This learning-based approach allows the algorithm to discover sophisticated, problem-specific strategies without manual intervention. Extensive evaluations on challenging combinatorial optimization problems demonstrate that AGE-MORL significantly outperforms state-of-the-art methods in both solution quality and stability.

We believe the proposed hierarchical-control paradigm is modular and can be integrated with a wide variety of population-based optimizers. Promising future directions include applying our framework to a broader spectrum of multi-objective optimization problems and investigating the use of a Large Language Model (LLM) as the high-level agent for more sophisticated strategic control.

## REPRODUCIBILITY STATEMENT

To ensure the reproducibility of our work, we provide detailed descriptions of our method and experiments. The AGE-MORL algorithm is formally described in Section 3 and presented as pseudocode in Algorithm 1. All experimental settings, including environment details, baseline configurations, and a full list of hyperparameters, can be found in Section 4 and Appendix A. We have provided the core implementation of our method and experiment scripts in the supplementary material. The full source code will be made publicly available upon publication.

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

## A  APPENDIX

This appendix provides comprehensive technical details for the Adaptive Guided Evolutionary Algorithm (AGE) framework designed for multi-objective vehicle routing problems. We present the complete state encoding methodology for meta-learning, mathematical formulations for performance metrics computation, detailed baseline algorithm implementations, experimental parameter configurations, and rigorous mathematical models for three problem variants: Traveling Salesman Problem (TSP), Capacitated Vehicle Routing Problem (CVRP), and Parallel Drone Scheduling TSP (PDSTSP). Furthermore, we include a sensitivity analysis for key hyperparameters and a disclosure on our use of LLMs in preparing this manuscript. All formulations are designed to support reproducible research and facilitate implementation of the proposed framework.

## A.1 LLM USAGE STATEMENT

During the preparation of this manuscript, we utilized a large language model (LLM) as a writing assistant. The LLM's role was strictly limited to improving the language and readability of our text. Specific uses included grammar correction, spelling checks, rephrasing sentences for clarity, and ensuring stylistic consistency. All scientific ideas, methodologies, experimental results, and conclusions presented in this paper were conceived and articulated by the human authors. The authors have carefully reviewed and edited the manuscript and take full responsibility for its final content.

## A.2 STATE ENCODING FOR META-LEARNING

The high-level agent observes the evolutionary algorithm's dynamic state through a 10-dimensional real-valued vector $\mathbf{s} \in \mathbb{R}^{10}$ to guide adaptive search strategy selection. The state encoding function $\phi : \mathcal{S} \to \mathbb{R}^{10}$ compresses the complex evolutionary state space $\mathcal{S}$ into normalized feature vectors for effective decision-making. The state vector $\mathbf{s} \in \mathbb{R}^{10}$ is defined in Table 2.

Table 2: Detailed definitions of the 10-dimensional state vector $s_t$.

| Symbol | Dimension Name | Detailed Description |
|--------|----------------|---------------------|
| $s_1$ | Iteration Progress | The ratio of the current iteration $t$ to the maximum number of iterations. |
| $s_2$ | Performance Level | The Hypervolume (HV) value of the current Pareto front, directly measuring solution quality. |
| $s_3$ | Improvement Trend | The average HV improvement rate over the last 5 iterations, capturing convergence speed. |
| $s_4$ | Convergence Degree | The ratio of the number of non-dominated solutions to the population size, $|PF_t|/N$. |
| $s_5$ | Distribution Diversity | The mean of standard deviations of the Pareto front across each objective, measuring solution spread. |
| $s_6$ | Coverage Extent | The mean of the range (max-min) of the Pareto front across each objective, indicating front extension. |
| $s_7$ | Weight Balance | The mean of standard deviations of the weight vectors, measuring the uniformity of search directions. |
| $s_8$ | Convergence Stability | The standard deviation of HV over the last 10 iterations, detecting search stability. |
| $s_9$ | Stagnation Detection | A binary indicator that is active when HV improvement is below a threshold. |
| $s_{10}$ | Dense Region Ratio | The proportion of solution pairs on the Pareto front with small pairwise distances. |

## A.3 PERFORMANCE METRICS COMPUTATION

### A.3.1 HYPERVOLUME INDICATOR

The hypervolume (HV) indicator measures the volume of objective space dominated by the Pareto front, simultaneously reflecting both convergence and distribution quality. Given Pareto front $A = \{a_1, \ldots, a_{|A|}\}$ and reference point $r$, the HV is calculated as:

$$\text{HV}(A, r) = \Lambda \left( \bigcup_{i=1}^{|A|} [a_i, r] \right) \tag{6}$$

where $\Lambda(\cdot)$ denotes the Lebesgue measure and $[a_i, r]$ represents a hyperrectangle. For problem instances with $n$ customers, the reference point is set to $r = (n, n)$, ensuring validity for normalized objective functions.

### A.3.2 INVERTED GENERATIONAL DISTANCE INDICATOR

The Inverted Generational Distance (IGD) indicator measures the average distance from reference front to approximation front, evaluating convergence quality and distribution uniformity. Given approximation front $A$ and reference front $Z^*$, the IGD is expressed as:

$$\text{IGD}(A, Z^*) = \frac{1}{|Z^*|} \sum_{i=1}^{|Z^*|} \min_{a_j \in A} \|z_i^* - a_j\|_2 \tag{7}$$

As for the reference front construction, we employ a combined reference front strategy:

1. Aggregate all algorithms' Pareto fronts: $S = \bigcup_k A_k$
2. Extract non-dominated solutions: $Z^* = \{s \in S \mid \nexists s' \prec s\}$

---

**Algorithm 2** IGD Calculation

---

**Require:** approximation_set (algorithm's approximated front)
**Require:** reference_front (pre-constructed reference front)
**Ensure:** IGD value
1: distances ← [ ]
2: **for** each ref_point ∈ reference_front **do**
3:    $d \leftarrow \min_{\text{approx\_point} \in \text{approximation\_set}} \|\text{ref\_point} - \text{approx\_point}\|_2$
4:    distances.append($d$)
5: **end for**
6: **return** mean(distances)

---

## A.4 BASELINE ALGORITHM IMPLEMENTATION

### A.4.1 PA2D ALGORITHM

PA2D-MORL is a multi-policy framework that employs a phased training strategy grounded in Pareto ascent direction decomposition theory. At its core, the method decomposes the multi-objective problem into several single-objective subproblems using a weighted aggregation of policy gradients:

$$\nabla_\theta J^{\pi_\theta}(\omega) = \sum_{i=1}^{m} \omega_i \nabla_\theta J_i^{\pi_\theta}, \quad \text{s.t.} \quad \sum_{i=1}^{m} \omega_i = 1 \tag{8}$$

By solving a minimum norm optimization problem, it computes a unified Pareto ascent direction $\alpha^*$ that guides non-Pareto optimal policies toward the front. This strategy is implemented through two key mechanisms: the Partitioned Greedy Randomized (PGR) policy selection, which partitions the objective space to select representative policies from all regions for training; and the Pareto Adaptive Fine-Tuning (PA-FT) mechanism, which identifies and fills gaps on the front by selecting policy pairs with the largest nearest-neighbor distances and updating them in opposite directions. For a detailed implementation and theoretical analysis, readers are referred to the original paper.

### A.4.2 MOEA/D ALGORITHM

MOEA/D is a classic algorithm that approaches multi-objective optimization by decomposing the problem into a collection of single-objective subproblems. Each of the $|P|$ subproblems is defined by a unique, uniformly distributed weight vector $\lambda_i$. These weight vectors are systematically generated, for instance, for a two-objective case, as:

$$\Lambda = \{\lambda_i = [w_1, w_2] \mid w_1 \in [0 : \delta\omega : 1], w_2 = 1 - w_1\} \tag{9}$$

where $\delta\omega$ is the weight interval. The core of MOEA/D lies in using a weighted aggregation function to convert the multi-objective problem into a scalar optimization task for each weight vector. A common choice is the linear weighted sum:

$$g(f(x), \lambda_i) = \sum_{j=1}^{m} \lambda_{ij} \cdot f_j(x) \tag{10}$$

At each iteration, the algorithm assigns the best-performing individual $\pi_i^*$ to each subproblem according to this scalarized objective:

$$\pi_i^* = \arg\max_{\pi \in P} g(f(\pi), \lambda_i) \tag{11}$$

The population is subsequently updated using Pareto dominance, where a new individual $\pi_{\text{new}}$ replaces an existing one $\pi_{\text{old}}$ if it dominates. To maintain diversity, inferior solutions may also be accepted with a probability $\rho_{sa}$.

### A.4.3  PARETO FRONT ADAPTION ALGORITHM

Pareto Front Adaption (PFA) is a method specifically designed for bi-objective optimization, which approximates the Pareto front through a time-dependent weight adjustment mechanism. The core of this approach is a normalized progress ratio function, $\delta(t)$, which quantifies the advancement of the search:

$$\delta(t) = \text{clip}\left(\frac{t - t_\rho}{T - t_\rho}, 0, 1\right) \tag{12}$$

where $t$ is the current iteration, $T$ is the total number of iterations, and $t_\rho$ is a warmup period. This progress ratio is then used to dynamically adjust the weight vectors for the entire population at each iteration:

$$w_1^{(t)} = w_1^{\text{base}} + \delta(t) \cdot \Delta\omega \tag{13}$$
$$w_2^{(t)} = 1 - w_1^{(t)} \tag{14}$$

Here, $w_1^{\text{base}}$ is a base weight, and $\Delta\omega$ is the adjustment step. This progressive weight update strategy guides the search to explore different regions of the front as time evolves. To further enhance exploration and prevent premature convergence, PFA also incorporates a random perturbation mechanism into the weight adjustment process. The algorithm typically employs a full population selection strategy, ensuring all individuals participate in the training to promote uniform optimization across the front.

### A.5  EXPERIMENTAL PARAMETER CONFIGURATION

Table 3: General Evolutionary Algorithm Parameters

| Parameter | Value | Description |
|---|---|---|
| Population Size ($N$) | 50 | Number of individuals |
| Maximum Generations ($T$) | 1000 | Termination condition |
| Number of Objectives ($m$) | 2 | Bi-objective optimization |
| Weight Range ($[\omega_{\min}, \omega_{\max}]$) | $[0.0, 1.0]$ | Linear weight boundaries |
| Adaptive Weight Step ($\delta\omega$) | 0.01 | Weight fine-tuning amplitude |
| Cooling Ratio ($\lambda_{\text{cool}}$) | 0.995 | Temperature decay factor |
| Simulated Annealing Prob. ($\rho_{sa}$) | 0.9 | Initial probability for accepting inferior solutions |

Table 4: Hyperparameters settings of high-level AGE-MORL agent

| Parameter | Default | Description |
|---|---|---|
| Exploration Rate ($\epsilon$) | 0.9 | $\epsilon$-greedy initial value |
| Exploration Decay Rate ($\epsilon_{\text{decay}}$) | 0.995 | Exploration rate decay |
| Minimum Exploration Rate ($\epsilon_{\min}$) | 0.05 | Exploration lower bound |
| State Dimension ($d_s$) | 10 | State encoding length |
| Hidden Layer Dimension ($d_h$) | 64 | Neural network width |
| Action Dimension ($d_a$) | 5 | Number of strategy selections |
| Experience Replay Capacity ($C_{\text{replay}}$) | 2000 | Buffer size |
| Batch Size ($B$) | 32 | Training batch size |

Table 5: Search Strategy Parameters

| Parameter | Default | Tuning Range | Description |
|---|---|---|---|
| Dense Region Radius Factor ($r_{\text{dense}}$) | 1.5 | $[1.0, 1.5, 2.0]$ | Dense search radius |
| Sparse Region Length Factor ($l_{\text{sparse}}$) | 2.0 | $[1.5, 2.0, 2.5]$ | Sector region length |
| Sparse Region Width Factor ($w_{\text{sparse}}$) | 0.8 | $[0.6, 0.8, 1.0]$ | Sector region width |
| Projection Mode ($\text{abs}_{\text{proj}}$) | False | $[\text{False}, \text{True}]$ | Absolute value projection switch |

Table 6: Problem Instance Configuration

| Problem Type | Costumers | Vehicles | Drones |
|---|---|---|---|
| TSP-50 | 50 | 1 | - |
| TSP-100 | 100 | 1 | - |
| TSP-200 | 200 | 1 | - |
| CVRP-50 | 50 | 2 | - |
| CVRP-100 | 100 | 3 | - |
| CVRP-200 | 200 | 5 | - |
| PDSTSP-50 | 50 | 1 | 2 |
| PDSTSP-100 | 100 | 1 | 3 |
| PDSTSP-200 | 200 | 1 | 4 |

## A.6 PROBLEM MATHEMATICAL MODELS

### A.6.1 TRAVELING SALESMAN PROBLEM (TSP)

**Problem Description.** Find the shortest path visiting all cities in a given city set and returning to the starting point. This study employs a bi-objective TSP variant considering two different distance measurement criteria.

**Mathematical Model.** Given city set $V = \{1, 2, \ldots, n\}$, distance matrices $D_1 = \{d_{ij}^1\}$ and $D_2 = \{d_{ij}^2\}$, decision variables:

$$x_{ij} \in \{0, 1\}, \quad \forall i, j \in V, i \neq j \tag{15}$$

**Constraints:**

$$\sum_{j \in V, j \neq i} x_{ij} = 1, \quad \forall i \in V \tag{16}$$

$$\sum_{i \in V, i \neq j} x_{ij} = 1, \quad \forall j \in V \tag{17}$$

$$\sum_{i \in S} \sum_{j \in S} x_{ij} \leq |S| - 1, \quad \forall S \subset V, |S| \geq 2 \tag{18}$$

where, equation equation 16 and equation equation 17 are the degree constraints, which ensure that each vertex is entered and exited exactly once. Equation equation 18 provides the subtour elimination constraints, preventing the formation of disconnected loops.

**Bi-objective Functions:**

$$f_1 = \sum_{i \in V} \sum_{j \in V, j \neq i} d_{ij}^1 \cdot x_{ij} \tag{19}$$

$$f_2 = \sum_{i \in V} \sum_{j \in V, j \neq i} d_{ij}^2 \cdot x_{ij} \tag{20}$$

The two distance metrics are calculated based on independent coordinate systems, creating inherent conflicts between different path length measures and establishing a genuine bi-objective optimization scenario.

### A.6.2 CAPACITATED VEHICLE ROUTING PROBLEM (CVRP)

**Problem Description.** Multiple capacity-constrained vehicles depart from a distribution center, serve all customers, and return, optimizing total travel distance and customer waiting time.

**Mathematical Model.** Given customer set $C = \{1, 2, \ldots, n\}$, distribution center $\{0\}$, vehicle set $K = \{1, 2, \ldots, m\}$, node set $V = C \cup \{0\}$, decision variables:

$$x_{ij}^k \in \{0, 1\}, \quad \forall i, j \in V, i \neq j, k \in K \tag{21}$$

$$t_i^k \geq 0, \quad \forall i \in C, k \in K \tag{22}$$

Here, $x_{ij}^k$ is a binary variable that equals 1 if vehicle $k$ travels directly from node $i$ to node $j$, and 0 otherwise. The variable $t_i^k$ represents the arrival time of vehicle $k$ at customer node $i$.

**Constraints:**

$$\sum_{k \in K} \sum_{j \in V, j \neq i} x_{ij}^k = 1, \quad \forall i \in C \tag{23}$$

$$\sum_{i \in C} q_i \cdot \left( \sum_{j \in V, j \neq i} x_{ij}^k \right) \leq Q, \quad \forall k \in K \tag{24}$$

$$\sum_{j \in C} x_{0j}^k \leq 1, \quad \forall k \in K \tag{25}$$

$$\sum_{i \in V, i \neq j} x_{ij}^k = \sum_{i \in V, i \neq j} x_{ji}^k, \quad \forall j \in C, k \in K \tag{26}$$

$$t_j^k \geq t_i^k + s_i + \tau_{ij} - M(1 - x_{ij}^k), \quad \forall i, j \in C, k \in K \tag{27}$$

These constraints collectively define a valid solution. Equation equation 23 ensures that each customer is visited exactly once. Equation equation 24 is the vehicle capacity constraint. Equation equation 25 ensures that each vehicle is used at most once, while equation equation 26 enforces flow conservation at each customer node. Finally, equation equation 27 sets the time constraints, which also serve to eliminate subtours. In these formulations, $q_i$ is the demand of customer $i$, $Q$ is the vehicle capacity, $s_i$ is the service time, $\tau_{ij}$ is the travel time, and $M$ is a sufficiently large constant.

**Bi-objective Functions:**

$$f_1 = \sum_{k \in K} \sum_{i \in V} \sum_{j \in V, j \neq i} d_{ij} \cdot x_{ij}^k \tag{28}$$

$$f_2 = \sum_{k \in K} \sum_{i \in C} t_i^k \tag{29}$$

These two objectives are often conflicting, as minimizing travel distance might lead to longer waiting times for some customers, and vice versa. The goal is to find a set of solutions that represent the best possible trade-offs between them.

### A.6.3 PARALLEL DRONE SCHEDULING TRAVELING SALESMAN PROBLEM (PDSTSP)

**Problem Description.** A single distribution center dispatches one truck and multiple drones for parallel operations. The truck serves customers along a TSP route while drones serve customers directly from the distribution center. A key feature of this model is that no synchronization is required between the truck and drones, allowing them to operate in parallel. This operational flexibility creates a direct trade-off between the two conflicting objectives: minimizing the total travel distance and minimizing the total mode-dependent operational cost.

**Mathematical Model.** Given customer set $C$, distribution center $\{0, c+1\}$, drone set $V$, decision variables:

$$\hat{x}_{ij} \in \{0, 1\}, \quad \forall i \in N_0, j \in \{N_+ : j \neq i\} \tag{30}$$

$$\hat{y}_{i,v} \in \{0, 1\}, \quad \forall i \in C'', v \in V \tag{31}$$

$$\hat{u}_i \in [1, c+2], \quad \forall i \in N_+ \tag{32}$$

Here, $\hat{x}_{ij}$ is a binary variable representing the truck routes. The variable $\hat{y}_{i,v}$ is also binary, indicating the drone service assignments, where $C'' \subseteq C'$ represents the set of customers reachable and serviceable by drones. Finally, $\hat{u}_i$ are auxiliary variables used for subtour elimination.

**Constraints:**

$$\sum_{i \in N_0, i \neq j} \hat{x}_{ij} + \sum_{v \in V} \hat{y}_{j,v} = 1, \quad \forall j \in C \tag{33}$$

$$\sum_{j \in N_+} \hat{x}_{0,j} = 1 \tag{34}$$

$$\sum_{i \in N_0} \hat{x}_{i,c+1} = 1 \tag{35}$$

$$\sum_{i \in N_0, i \neq j} \hat{x}_{ij} = \sum_{k \in N_+, k \neq j} \hat{x}_{j,k}, \quad \forall j \in C \tag{36}$$

$$\hat{u}_i - \hat{u}_j + 1 \leq (c+2)(1 - \hat{x}_{ij}), \quad \forall i \in C, j \in \{N_+ : j \neq i\} \tag{37}$$

$$1 \leq \hat{u}_i \leq c+2, \quad \forall i \in N_+ \tag{38}$$

$$\hat{x}_{ij} \in \{0,1\}, \quad \forall i \in N_0, j \in \{N_+ : j \neq i\} \tag{39}$$

$$\hat{y}_{i,v} \in \{0,1\}, \quad \forall i \in C'', v \in V \tag{40}$$

Equation equation 33 ensures that each customer is serviced exactly once, either by the truck or a drone. Equations equation 34 and equation 35 manage the truck's departure from and return to the depot, while equation 36 enforces flow conservation for the truck's route. The subtour elimination for the truck's path is handled by equations equation 37 and equation 38 using auxiliary variables. Finally, equations equation 39 and equation 40 define the binary nature of the truck routing and drone assignment decision variables, respectively.

**Bi-objective Functions:**

$$f_1 = \sum_{i \in N_0} \sum_{j \in N_+, j \neq i} d_{ij} \cdot \hat{x}_{ij} + \sum_{v \in V} \sum_{i \in C''} d'_{i,v} \cdot \hat{y}_{i,v} \tag{41}$$

$$f_2 = \sum_{i \in N_0} \sum_{j \in N_+, j \neq i} c^t_{ij} \cdot \hat{x}_{ij} + \sum_{v \in V} \sum_{i \in C''} c^d_{i,v} \cdot \hat{y}_{i,v} \tag{42}$$

Here, the first objective function is to minimize the total travel distance, which includes the paths of both the truck and all serving drones. $d_{ij}$ and $d'_{i,v}$ denote travel distances for trucks and drones respectively. The second objective aims to minimize the total operational cost, calculated from the mode-dependent costs of the truck and drones. $c^t_{ij}$ and $c^d_{i,v}$ represent operational cost rates for trucks and drones respectively. The problem exhibits inherent trade-offs: drones incur longer round-trip distances to the distribution center but have lower operational costs, creating genuine conflicts between distance minimization and cost optimization objectives.

## A.7 PARAMETER ANALYSIS

We analyze the sensitivity of key hyperparameters within the AGE-MORL framework, with experiments conducted on the TSP and CVRP environments with 30 nodes. The parameters studied include the high-level agent's learning rate $lr$ and discount factor $\gamma$, as well as the geometric search parameters: the dense region radius factor $r$, and the sparse sector's length $l$ and width $w$ factors. The results are summarized in Table 7.

The agent's learning parameters, $lr$ and $\gamma$, exhibit some sensitivity to the problem domain. As shown in the table, no single value is optimal for both environments, suggesting that these parameters may benefit from minor tuning depending on the task. However, moderate values such as $lr = 0.001$ and $\gamma = 0.9$ generally provide a robust performance baseline. For the geometric search parameters, a larger dense search radius ($r = 2.0$) tends to improve performance by allowing for more thorough local refinement. For the sparse sector search, a comprehensive Full Scan (FS) is generally more effective than a Half Scan (HS), and a larger search sector (e.g., $l = 2.5, w = 1.0$) often yields the best results, indicating that aggressive exploration in sparse regions is beneficial.

Table 7: Hyperparameter analysis results on TSP-30 and CVRP-30 environments.

| H | Values | TSP$_{30}$ | | CVRP$_{30}$ | |
|---|---|---|---|---|---|
| | | HV ↑ | IGD ↓ | HV ↑ | IGD ↓ |
| $lr$ | 0.001 | 707.9436 | 0.6372 | 795.5598 | 0.9431 |
| | 0.005 | 686.1736 | 0.5764 | 805.2762 | 1.3914 |
| | 0.01 | 665.9665 | 0.8681 | 823.0586 | 0.7973 |
| $\gamma$ | 0.9 | 727.1849 | 0.6446 | 758.1754 | 1.0401 |
| | 0.95 | 709.3930 | 0.8954 | 771.5468 | 0.5061 |
| | 0.99 | 704.4317 | 1.3464 | 777.1132 | 0.7461 |
| $r$ | 1.0 | 706.7581 | 1.7155 | 782.2927 | 1.3894 |
| | 1.5 | 710.8975 | 0.8065 | 790.9394 | 1.0115 |
| | 2.0 | 730.8973 | 0.9005 | 798.3276 | 0.6659 |
| $l, w$ | 1.5,0.6(HS) | 691.5902 | 1.0191 | 790.3732 | 1.2113 |
| | 2.0,0.8(HS) | 678.0151 | 1.0445 | 817.9426 | 0.9429 |
| | 2.5,1.0(HS) | 723.2910 | 0.6509 | 846.5363 | 0.5581 |
| | 1.5, 0.6(FS) | 697.7071 | 0.7504 | 772.5354 | 1.3326 |
| | 2.0, 0.8(FS) | 697.3640 | 0.9359 | 834.7988 | 0.9053 |
| | 2.5, 1.0(FS) | 718.9769 | 0.7489 | 849.8688 | 0.9143 |

