# OpenReview forum: "AGE-MORL: Agent-Guided Evolutionary Control for Multi-Objective Reinforcement Learning"
_ICLR.cc/2026/Conference — Submitted to ICLR 2026_

### Official Review · Reviewer_6rPa · 2025-10-26

**Soundness:** 3
**Presentation:** 3
**Contribution:** 3
**Rating:** 6
**Confidence:** 4

**Summary:**

This paper proposes AGE-MORL, a hierarchical framework that uses a high-level Deep RL agent to adaptively select search operators for multi-objective evolutionary algorithms. The operator selection is formulated as an MDP, where the agent learns dynamic policies based on the optimization state. The framework incorporates geometry-aware search operators that analyze the Pareto front to distinguish dense and sparse regions, plus a Blink mechanism with adaptive Simulated Annealing for diversity preservation. Experiments on multi-objective combinatorial problems (TSP, CVRP, PDSTSP) show AGE-MORL significantly outperforms baselines (PA2D-MORL, MOEA/D, PFA) in solution quality and stability across problem scales.

**Strengths:**

The paper presents a **well-motivated approach** addressing static search strategies in multi-objective evolutionary algorithms through a clear **hierarchical MDP formulation**. The **geometry-aware operators** and synergistic mechanisms (Blink + adaptive SA) show thoughtful design for MORL challenges. **Experimental validation is solid**, covering multiple problems (TSP, CVRP, PDSTSP) at various scales with consistent improvements over baselines in hypervolume and IGD metrics. The **ablation study** effectively demonstrates the value of adaptive agent control, and the paper is **clearly written** with good reproducibility.

**Weaknesses:**

The paper has several notable weaknesses.
The framework introduces many hyperparameters (learning rates, temperature schedules, geometric thresholds c_d, c_s, radius factor r, sector factors l and w, Blink probabilities), creating significant tuning complexity that undermines claims of reducing manual intervention.
The geometry-aware operators rely on simplistic assumptions—defining dense/sparse regions using only pairwise distances with fixed thresholds may be insufficient for complex, non-uniform Pareto fronts with irregular geometries.
Computational efficiency is not discussed: the overhead of training the high-level agent, computing geometric features, and maintaining the MDP state at each iteration could be substantial, yet no runtime comparisons or scalability analysis are provided.
The experimental scope is limited to combinatorial routing problems with similar structure; generalization to other MORL domains (continuous control, resource allocation) remains unvalidated.
Finally, the baseline comparisons exclude recent learning-based MORL methods, and the improvement margins, while consistent, are often modest relative to the added complexity.

**Questions:**

no more.

---

### Official Review · Reviewer_bV9k · 2025-10-28

**Soundness:** 3
**Presentation:** 1
**Contribution:** 2
**Rating:** 4
**Confidence:** 5

**Summary:**

The paper introduces AGE-MORL, a novel hierarchical framework aimed at enhancing multi-objective reinforcement learning (MORL) through adaptive search operator selection. By employing deep reinforcement learning (DRL) as a high-level controller, AGE-MORL dynamically balances exploration and exploitation in the search process. The framework also incorporates geometry-aware search operators to target specific regions of the Pareto front and combines a Blink mechanism with an adaptive Simulated Annealing (SA) criterion to prevent premature convergence. Evaluated on several multi-objective combinatorial optimization problems, AGE-MORL demonstrates superior performance in solution quality and stability compared to baseline methods.

**Strengths:**

1. The AGE-MORL framework effectively combines a high-level DRL agent with a low-level MOEA, enabling dynamic and adaptive selection of search operators. This addresses the limitations of traditional methods that rely on static and non-adaptive strategies.

2. The design of intelligent search operators based on geometric analysis of the Pareto front allows for targeted exploration and exploitation, leading to more efficient and effective optimization.

3. The integration of the Blink mechanism and adaptive SA criterion helps maintain population diversity and enables the algorithm to escape local optima, thereby enhancing the robustness of the search process.

4. The paper provides extensive experimental results across diverse multi-objective optimization problems, demonstrating AGE-MORL's superiority over baseline methods in terms of effectiveness.

**Weaknesses:**

1. The hierarchical structure and integration of multiple mechanisms, including the DRL agent, geometry-aware operators, Blink mechanism, and SA criterion, may increase the complexity of the framework. This could result in higher computational overhead and pose challenges in implementation and comprehension.

2. Although the method shows promise in the tested combinatorial optimization problems, its effectiveness in other types of multi-objective optimization problems (especially problems with >=3 objectives) remains to be validated. The geometric analysis and operators might require adjustments for different problem domains.

3. The performance of AGE-MORL could be sensitive to hyperparameter selection, such as learning rates, discount factors, and geometric search parameters. This may necessitate careful tuning for optimal performance across different scenarios.

4. The presentation of Section 3.2 and Section 3.3 could be enhanced with additional details.

5. The paper does not compare its method with other state-of-the-art neural methods.

6. The paper contains several grammatical errors, incomplete sentences, and missing definitions, which impact its readability. Furthermore, Figure 2 is not referenced in the main text.

**Questions:**

1. What are the computational requirements and efficiency of AGE-MORL compared to other methods? The paper emphasizes solution quality and stability but does not provide a detailed analysis of running time.

2. What is the architecture of the Q-network? Does the design of the Q-network significantly impact the results?

3. Can AGE-MORL be integrated with other optimization algorithms or frameworks? The paper suggests potential for integration with various population-based optimizers, but specific strategies and implications for such integrations are not discussed.

4. How does the performance of AGE-MORL scale with larger and more complex problems? The experimental evaluation covers problems with up to 200 nodes, but further analysis on larger-scale instances would provide better insights into the method's generalization ability.

5. What are the limitations of the geometry-aware search operators in accurately identifying and targeting different regions of the Pareto front? The effectiveness of these operators may vary depending on the problem's characteristics and the quality of the geometric analysis.

6. How does the Blink mechanism interact with the adaptive SA criterion in practice? A more detailed ablation study of their synergistic effect and how they influence each other during the search process would enhance the understanding of the framework's dynamics.

7. How does AGE-MORL handle problems with more than two objectives? The paper primarily focuses on bi-objective optimization problems. Extending the method to many-objective problems may introduce additional challenges in Pareto front analysis and operator selection.

---

### Official Review · Reviewer_Bmwq · 2025-10-28

**Soundness:** 2
**Presentation:** 2
**Contribution:** 2
**Rating:** 2
**Confidence:** 4

**Summary:**

This work presents a framework that employs RL to adaptively select operators for multi-objective optimization algorithms within a formulated MDP. The approach integrates techniques such as sector-based exploration and a blink mechanism to enhance the quality of decision-making.

**Strengths:**

1. Despite potential stability issues with the high-level RL agent, the ablation results indicate some effectiveness.

2. The overall framework design is generally reasonable.

**Weaknesses:**

1. The low-level RL algorithm is not well explained. It is unclear how it explores when receiving a high-level action. Does it optimize a scalarized objective with an exploratory weight?

2. With a 10-dimensional state space and only a few hundred high-level transitions, the method is likely far from convergence, given the poor sample efficiency of deep RL. The observed improvement appears to stem mainly from the MDP design, in which almost no poor decisions can be made.

3. Experiments are limited to discrete problems. The approach may struggle in continuous control tasks, where similar high-level state representations could correspond to different policy sets, potentially omitting important information for effective high-level decision-making.

4. The work is largely engineering-oriented and lacks conceptual or theoretical insights.

**Questions:**

1. See weekness 1
2. How big is the neural network for high level agent

---

### Official Review · Reviewer_ofuK · 2025-10-29

**Soundness:** 2
**Presentation:** 1
**Contribution:** 2
**Rating:** 2
**Confidence:** 5

**Summary:**

This paper proposes using a RL–based policy to dynamically select search operators for refining solutions in multi-objective evolutionary algorithms (MOEAs). By employing an RL-based mechanism for operator selection, the method aims to mitigate premature convergence and avoid static strategies that neglect the current state of the Pareto front. The authors empirically demonstrate the effectiveness of their approach on various multi-objective combinatorial optimization (MOCO) problems. **However, the paper appears to violate the formatting template by reducing the original page margins.**

**Strengths:**

1. The paper presents a sound idea that leverages an RL-based policy to adaptively select operators within MOEAs.

2. Empirical results on MOCO problems demonstrate the effectiveness of the proposed method.

**Weaknesses:**

1. The paper is not well-organized or polished and appears not to have been carefully checked before submission. Specifically, the authors have altered the paper template by reducing the page margins. In addition, the abstract begins with an incomplete sentence that should be removed. Furthermore, the overall pipeline is not clearly described, and insufficient emphasis is placed on the main contributions of the work.

2. The paper frequently references multi-objective reinforcement learning (MORL), but the proposed method seems to operate purely within the MOEA setting, with no direct connection to existing MORL algorithms from a technical standpoint. It would be more appropriate to introduce MOEA rather than MORL in the preliminaries.

3. The method is evaluated only on MOCO problems and is not tested on more general MORL tasks, which limits the scope of the experimental validation.

4. The selected baseline methods are relatively weak and outdated. Incorporating more recent and competitive baselines would improve the credibility of the experimental comparison.

**Questions:**

1. Could the authors elaborate on the specific design of the action space in their proposed method?

---

### Meta-Review · Area_Chair_v4Xh · 2025-12-30

**Summary:**

In this paper, the authors considered multi-objective RL and proposed AGE-MORL based on a evolutionary  framework combined with simulated annealing for multi-policy MORL to prevent premature convergence and enlarge the HV of Pareto front.  The main idea is to adopt an DRL agent to adaptively select search operator for the evoluationary framework and reward design for this.   Experiments show some promising results.
It is recommened that the authors incorporate the reviewers' comments and improve the paper for possible future submission.

**Reviewer Concerns:**

1. The paper is not carefully proof-read with broken sentences and wrong fomat of citations.
2. The lower-level RL polices are not well described and it is unclear how the higher-level DQN algorithm interacts with the lower-level RL population. Thus, the paper seems incomplete.
3. Experiments are limited with rather old baselines.

**Reviewer Scores:**

Reviewer ofuK: 2,
Reviewer Bmwq: 2,
Reviewer bV9k: 4,
Reviewer 6rPa: 6

The authors did not provide their rebuttal. So, the score will not change.

---

### Decision · Program_Chairs · 2026-01-26

Reject